# Role of health facility governing committees in strengthening social accountability to improve the health system in Tanzania: protocol for a participatory action research study

Miguel San Sebastian ![ORCID],[1] Stephen Maluka ![ORCID],[2] Peter Kamuzora,[3] Ntuli A Kapologwe,[4] Ramadhani Kigume,[3] Cresencia Masawe,[2] Anna-Karin Hurtig[1]

¹Department for Epidemiology and Global Health, Umeå University, Umeå, Sweden
²Dar es Salaam University College of Education, Dar es Salaam, Tanzania
³Institute of Development Studies, University of Dar es Salaam, Dar es Salaam, Tanzania
⁴President's Office Regional Administration and Local Government, Dodoma, Tanzania

**Correspondence to**
Dr Miguel San Sebastian;
miguel.san.sebastian@umu.se

## ABSTRACT

**Introduction** Social accountability is important for improving the delivery of health services and empowering citizens. The government of Tanzania has transferred authority to plan, budget and manage financial resources to the lower health facilities since 2017. Health facility governing committees (HFGCs) therefore play a pivotal role in ensuring social accountability. While HFGCs serve as bridges between health facilities and their communities, efforts need to be made to reinforce their capacity. This project therefore aims to understand whether, how and under what conditions informed and competent HFGCs improve social accountability.

**Methods and analysis** This study adopts a participatory approach to realist evaluation, engaging members of the HFGCs, health managers and providers and community leaders to: (1) map the challenges and opportunities of the current reform, (2) develop an initial programme theory that proposes a plan to strengthen the role of the HFGCs, (3) test the programme theory by developing a plan of action, (4) refine the programme theory through multiple cycles of participatory learning and (5) propose a set of recommendations to guide processes to strengthen social accountability in the Tanzanian health system. This project is part of an ongoing strong collaboration between the University of Dar es Salaam (Tanzania), and Umeå University (Sweden), providing opportunities for action learning and close interactions between researchers, decision-makers and practitioners.

**Ethics and dissemination** Ethical approval to conduct the study was obtained from the National Ethical Review Committee in Tanzania— National Institute for Medical Research (NIMR/HQ/R.8a/Vol.IX/3928). Permissions to conduct the study in the health facilities were given by the President's Office Regional Administration and Local Government and relevant regional and district authorities. The results will be published in open-access, peer-reviewed journals and presented at scientific conferences.

## STRENGTHS AND LIMITATIONS OF THIS STUDY

⇒ The study methodology is novel by adopting community-based participatory research theory-driven evaluation combining realist evaluation (RE) and soft systems methodology (SSM).
⇒ The findings will be context-bound and intervention might be labour intensive.
⇒ The process of integrating RE and SSM is not detailed in the literature and will therefore require high level of flexibility and skilled facilitation.
⇒ Dissemination of the findings in understandable language to the different levels of the district health system (from dispensary to the hospital) can be challenging.

## INTRODUCTION

Over the past two decades, there has been a growing consensus that social accountability mechanisms are important for improving the delivery of public services and empowering citizens, particularly in resource poor settings.[1–3] Social accountability refers to the ability of civil society and citizens to hold public officials and public service providers accountable for the provision of services, including health services. Social accountability has emerged as an important means of strengthening community participation and community level systems, and as a key link between community users and public service providers.[4] Specifically in the area of health systems, social accountability is gaining rapid acceptance as a way to address health system inefficiencies and improve basic public health performance, including planning and service delivery and to attain the highest possible standards of health.[5 6] In addition, strengthening social accountability initiatives has the potential to redress the causes of health inequalities and promote better governance of health systems. These initiatives put the users of services at the frontline of accountability, enabling those affected to advocate for change.[7 8] Many initiatives have

emerged over the last decade that support citizens to act collectively and engage with state authorities to demand accountability in the health system, as well as other public sectors. These initiatives tend to be highly heterogeneous due to the variety of strategies they employ and the role of contextual conditions, such as power relations and political opportunity, in shaping implementation and impact. Positive results have been demonstrated in areas including improved user satisfaction, citizen and community empowerment, reduction in informal charges and increased resource allocation for drugs, although broader structural changes have also proven more resistant to these citizen-led initiatives.[3 9–11]

As in many other low-income and middle-income countries (LMICs), in Tanzania, health facility governing committees (HFGCs), consisting of community members, have become integral in decentralisation through devolution reforms and a key intervention promoting social accountability within the health system. HFGCs are expected to help healthcare providers make decisions that best serve the interests of the community. The Tanzanian health system assumes a pyramid shape, where the base represents the community the HFGCs are coming from. The HFGCs are to approve all transactions made at the health facilities, and they play pivotal roles in the inspection of health commodities that arrive at the facilities prior to dispensing. They are also part of the planning and budgeting team for the health facility and supervise minor renovations or constructions of the facility.[12–14] Fiscal decentralisation has been advocated as very important in delivering autonomy to decentralised health levels in a decentralised health service delivery.[15] It is expected that by shifting fiscal authorities to subnational levels such as HFGCs, the delivery of health services will fulfil local demands and respond to community needs on time. Advocates of performance-based financing have put the HFGCs at the core of health reform, expecting this to hold healthcare providers accountable for the performance of their health facilities.[16] Under the current arrangements, each health facility (dispensaries, health centres and council hospitals) should prepare its comprehensive annual health plan and budget, and funds for its implementation are transferred directly from the Ministry of Finance and Planning to the health facilities' accounts; the policy is known as Direct Health Facility Financing (DHFF).[17]

Recent studies in Tanzania have explored the functionality of the HFGCs, recognising the poor to average performance, that they are largely inactive and rarely provide an oversight function, and that their involvement in providing governance is limited.[12 14 18–20] The key bottlenecks often include limited skills and capacity, limited awareness of the community members regarding the roles and responsibilities of the HFGCs, inadequate interest in and support among key health workers or managers, lack of interest from communities, inadequate access to financial resources and a lack of financial incentives for the committee members and thus difficulty in discharging and sustaining voluntary membership over time.[12 18 21] Health facility planning and budgeting is consequently ad hoc and not evidence based. Despite these weaknesses, the government of Tanzania has further transferred power and authority to plan, budget and manage financial resources to the health facilities since 2017.[14]

Explorative studies from different contexts have suggested that training[11 18 19] is a way forward to empower HFGCs, but they have also highlighted the importance of paying attention to the socioeconomic and cultural contexts in which they operate, including power relationships between healthcare providers and the HFGCs, and political capabilities and how HFGCs perceive their role.[8 19 20] There is, however, a scarcity of participatory (action) research on *how* to better strengthen and implement social accountability mechanisms, such as the HFGCs, and their relationships with healthcare providers and the community they serve.[16 18 22]

In this paper, we present the protocol of a project which has the overall aim of understanding whether, how and under what conditions informed and competent HFGCs can strengthen social accountability. There are three specific research objectives:

1. To explore stakeholder experiences and perceptions of how HFGCs contribute (or not) to social accountability in the local health system, and identify the underlying causes.
2. To strengthen the ability of HFGCs to perform their roles in promoting social accountability through participatory learning.
3. To contribute to methodological development in the co-production of knowledge.

The project has a duration of 3 years, starting in late 2022. At the time of preparation of the current protocol, the project is at the initial stages of data collection.

## METHODS AND ANALYSIS

These aims will be achieved by combining realist evaluation (RE) and participatory methods (soft system methodology) and by engaging members of the HFGCs, health managers and providers and community leaders to (1) map and analyse the challenges and opportunities of the current reform in transferring power, authority and autonomy to the health facilities; (2) develop an initial programme theory that proposes a plan to strengthen the ability of the HFGCs to perform their tasks in line with the roles outlined in the current policy; (3) test the programme theory by developing a plan of action; (4) refine the programme theory through multiple cycles of participatory learning; and (5) propose a set of recommendations which will provide useful lessons to guide processes and actions to strengthen social accountability in the Tanzanian health system.

### Study setting

Like other LMICs, Tanzania has been implementing health sector decentralisation since the 1990s, when the

authority and responsibility for planning and managing district health services were transferred from the Ministry of Health to the District Councils across the country. The country has 6074 public primary health facilities, of which 184 are District Councils, 650 are health centres and 5240 are dispensaries.

Since the inception of health-sector reforms in Tanzania in the 1990s, health planning and its implementation have been decentralised to the Council Health Management Teams (CHMTs) headed by the District Medical Officer (DMO). The CHMTs were responsible for managing local authority funds disbursed from the central government. Each District Council was required to prepare an annual Comprehensive Council Health Plan, showing all the priorities of the district, including those of the health facilities, and the funds that have been allocated to address each priority area and individual cost centres, that is, the DMO, council hospital, health centre, dispensary and community.[21] This decentralisation arrangement centralised autonomy for planning, budgeting and resource allocation at the district level, and health facilities and the community had limited decision-making space and authority. Consequently, in most cases, health facilities were often cash-strapped due to the inadequate and late receipt of funds from the District Council, which ultimately hindered the effective and efficient delivery of healthcare services. Funds at the District Council level were also misappropriated, and on occasion, District Councils were liable for fraud. In 2017, the government of Tanzania, through the introduction of the DHFF approach, transferred authority to plan, manage and make decisions to the subdistrict and primary healthcare facilities.[14 23] Through DHFF, funds for financing health services are transferred directly from the central government through the Ministry of Finance and Planning to the public primary healthcare facilities. Similarly, funds generated through user fees (out-of-pocket fees), health insurance schemes and other local (council owned) sources are managed at the health facility level by the HFGCs. HFGCs thus have considerable autonomy in planning, budgeting and managing funds to ensure the delivery of healthcare services.

## Study design

This research project will adopt an action research approach labelled community-based participatory research theory-driven evaluation, combining RE[24] and soft systems methodology (SSM)[25] as suggested by Dalkin *et al*.[26]

RE is a theory-driven approach that focuses on explaining the mechanisms of action that underlie a complex programme or intervention. Just like SSM, RE is underpinned by complex theory thinking. The SSM has an explicit orientation towards the experiential learning of a group of participants, and therefore adds value to the 'method neutral' RE by engaging stakeholders from the outset. This combination responds to a recent call that REs should adopt participatory (action) research

methods and engage with issues of (power) inequities contributing to 'a better understanding of the intervention and may facilitate the emancipation of the disenfranchised'.[26 27] The combination of RE and SSM allows for inclusion of local stakeholders which will support complementary theory building and theories being rooted in lived realities.

Below we start by defining RE and SSM separately, and then later we describe how we will combine both approaches in practice.

### Realist Evaluation

RE has proven to be useful when exploring complex health interventions. The intervention in our study is the 2017 national reform which transferred power, autonomy and authority to plan, budget and manage financial resources to the HFGCs in order to ensure social accountability. RE aims to ascertain why, how and under which circumstances interventions succeed or fail. Grounded in scientific realism and based on the work of Pawson and Tilley, RE focuses on how the mechanisms of change are triggered by the combination of the intervention and contextual factors, which, in turn, leads to intended (or unintended) outcomes. RE begins by making the programme theory underlying the development of an intervention explicit. The programme theory is understood as every day, prosaic theories that explain how social problems are generated and how interventions can help to solve them.[24] The programme theory is afterwards tested through the observation of experiences in real cases where the intervention has been implemented. Data collected in these cases are used to refine the preliminary programme theory and specify a middle-range theory, thereby providing plausible explanations of why, how and under what circumstances the intervention triggered mechanisms that resulted in certain outcomes. RE can, as in our case, offer an opportunity to adapt, refine and contextualise an intervention. The data collected will not only help to ascertain the programme theory, but will also inform the adaptation of the interventions and guide the evaluation. RE has shown to be a useful way to bridge the gap between theory building and practical recommendations;[28] if we are able to identify the mechanisms that lead to positive change, they can guide scaling-up processes.

### SSM: a community-based participatory research approach

The SSM is based on systems theory, which attempts to take a holistic view of the interrelations of component parts, 'the wider picture'.[25] Like many other systems approaches, SSM encourages exploration and problem solving through a stepwise process that allows for distinctions to be drawn between the 'real world' and 'systems thinking about the real world'. The 'real world' is the world where the problem occurs and the human activity that takes place within it. The 'systems world' is the context of analysis in which the information from the real world is scrutinised and dissected in the problem-solving process. SSM is an action-oriented process of enquiry into

situations in which users learn their way from finding out about the situation, to taking action to improve it. SSM uses maps to explore and questions the key structures, processes, people, issues expressed by stakeholders and conflicts in the programme, as well as its broader social, cultural and political contexts.[25][29] The systems maps created are set against perceptions of the real world by a process of comparison that initiates debate between stakeholders. As a result of these comparisons, stakeholders gain a better understanding of a situation and are engaged in reflexive learning cycles. SSM thus operationalises the co-construction of social reality maps with the greatest face validity for the system actors themselves. In order to ensure theoretical validity, SSM relies on the views of those on the 'inside' of a situation, exploring with them their perceptions of system structures; how their activity is reflected in these structures and how they would like things to change.[29]

### Stepwise methods

Drawing on the methodology proposed by Dalkin *el al*,[26] we will merge RE and SSM. We will follow a stepwise approach similar to the 'classical' RE which includes situating and understanding the intervention in context, developing the preliminary programme theory, testing the programme theory and refining it.[28] We will embed SSM in these steps to guide the understanding of the situation, action planning, evaluation and specification of the learning achieved in a participatory co-learning process.

This research project will be implemented in two districts in order to examine variations. These districts will be purposively selected based on their performance in the 2018 Star Rating Assessment conducted by the Ministry of Health,[30] choosing one high and one poor performer. Other criteria will include geographical balancing of the regions in the zones, as well as a balance between rural and urban districts. In each district, the project will be carried out together with the HFGCs and other stakeholders related to one hospital, two health centres and four dispensaries. This sampling is based on our experience, which shows that most districts in Tanzania follow this structure.

### Step 1: mapping complexity: understanding social accountability through competent HFGCs in their context

In this stage, the problem situation is explored and expressed in a first SSM map. The object of this stage is to display the situation of the HFGC and the role the HFGC plays in promoting social accountability.

A 1-day workshop with each of the HFGCs, representatives of the communities and health providers will be held in each of the two study districts. An introduction to the role of the committees and the rights and duties of its members will first be provided. After that, a discussion in small groups will be conducted to explore the challenges and opportunities of HFGCs to improve social accountability in the local health system. SSM highlights certain generic issues that should be included in the discussions,

such as structures, processes, climate, people and conflicts. As recommended by the method, we will summarise the output of the discussions, drawing a map which shows the relationships and complexity of the system. This activity will be supplemented by the non-participatory observation of some meetings for all HFGCs involved.

### Step 2: understanding complexity: developing the initial programme theory

The objective of this stage is to create an initial programme theory to explain how the HFGCs will contribute to social accountability in the context.

The SSM map from Step 1 will form the point of departure for this step. A root definition is used to think about what is likely to 'make sense' to address the problem situation, and as a step in the development of a conceptual model. It will encompass a number of elements that Checkland characterised under the mnemonic 'CATWOE' (customers, actors, transformation process, 'world view', owners and environmental constraints).[25] Using these elements (after adapting them to the context), a root definition will be formulated in order to express the core purpose and activity of the HFGCs. In this step, a second map will depict how the HFGCs should work, providing the potential mechanisms and contexts. This will form the basis for the formulation of the initial programme theory. Researchers will discuss the different root definitions leading to different programme theories, and thus reach a visualisation by consensus in an SSM Map 2 to be presented to the HFGCs and other stakeholders in the next step.

### Step 3: planning for actions: testing the initial programme theory

This step will combine the action-taking and evaluation phases of the SSM approach and will provide an opportunity to test the initial programme theory (or theories).

Once the initial programme theory has been visualised in SSM Map 2, it will be time to compare it with the real-world model (SSM Map 1) from Step 1. This will be done through a participatory approach gathering again representatives of the HFGCs, communities and health providers in a 2-day workshop. The workshop will include three parts. In the first, we will adopt a matrix approach to compare the initial programme theory to the real-world model (SSM Map 1). The matrix looks at each component of the model and poses the following questions: (1) what similarities and differences are there between initial programme theory and the 'real world' (SSM Map 1); (2) how do these factors affect the role of the HFGCs; and (3) what strengths could be built on to achieve change? The output of this first part of the workshop is a matrix including activities, their achievement and how they are carried out.

In the second part, the participants will discuss what changes could bring about improvements in the system regarding the three dimensions that the policy highlights as the responsibility of the HFGCs: planning, budgeting and management. Feasibility and desirability will be

considered so that any change is discussed and implemented with the agreement of participants, paying full regard to the culture, environment and politics of the contextual system.

The final part will be concerned with the implementation of changes to address the problem. To ensure that these will be implemented in a clear and measurable way, the planned changes will be translated into specific targets, desired outcomes and persons responsible for the target within defined time boundaries.

### Step 4: implementing actions: refining the programme theory

This step uses the findings from previous steps to enhance the understanding of how, why, for whom and under what circumstances HFGCs contribute to social accountability in the health system. The plan of action (see Step 3) is intended to build competent and informed HFGC members for the improvement of social accountability. We will use a multidisciplinary approach, employing the diverse skills of our research team, to finalise the co-development of the learning and actions aimed to strengthen capacity and accountability mechanisms.

Every 6 months the lessons and achievements will be discussed and the next cycle refined using SSM maps. The cyclic nature of the SSM allows the situation to be revised and reconstructed in light of the current situation and provides opportunities for learning.

Outcome measures to assess impact will be defined in a participatory manner through the different cycles of the project. Performance indicators from the Star Rating Assessment, which is conducted every 2 years, for the selected districts will also be used to monitor social accountability changes.

During this phase, lessons drawn will be used to better understand how to improve competencies among HFGCs and the local health governance system for social accountability. The lessons will include an account of 'what worked for whom in the context of the districts'. The lessons (mechanisms that lead to positive change) will be applied to modify HFGCs' induction and integration into the health system. Final workshops will be conducted with different stakeholders at national and district levels to reflect on and disseminate the findings of the study.

### Patient and public involvement

This study will not involve patients. However, given the participatory nature of the project, different stakeholders will be included in the data collection stage and in the discussion and dissemination of the findings.

### ETHICS AND DISSEMINATION

Several ethical dilemmas may arise when conducting participatory-related research, connected to informed consent and confidentiality. At the beginning of the research process, a memorandum of understanding that clearly delineated how we were going to handle data collection, reporting and dissemination was developed.

When data collection started, in late 2022, all study participants were informed of their rights and risks of participating in the study. To that end, oral consent is being obtained from all those participating in the study at the village level. Verbal consent has been preferred because in the villages signing written consent would scare respondents from participating in the study. Written consent will, however, be used when including health workers and policymakers. More importantly, throughout this study, privacy and confidentiality are ensured, as personal details of the participants are not linked with the provided information. Likewise, access to the supplied information is limited to interviewers and core team members.

Research ethics approval has been obtained from the National Ethical Review Committee in Tanzania—National Institute for Medical Research (NIMR) with certificate no. NIMR/HQ/R.8a/Vol.IX/3928. Permissions have been given by the President's Office Regional Administration and Local Government and relevant regional and district authorities to conduct the study in their health facilities (hospitals, health centres and dispensaries).

The overall approach that this research project will adopt is the engagement of the decision-makers at the national, district and community levels. At the national level, the project will work with two central government ministries whose responsibilities relate directly to the district health systems in Tanzania. At the district level, the project will focus on two important bodies: the District Council and the CHMT. The District Council is a decision-making body that oversees all the development activities in the district. The CHMT is responsible for the planning and management of the district health services and implementation of policies.

The results will be published in open-access, peer-reviewed journals and presented at scientific conferences.

### DISCUSSION

In sub-Saharan Africa, and Tanzania specifically, concerns have been raised regarding the access to, quality of and effectiveness of health services, particularly for the most vulnerable populations. Strong and well-functioning health systems not only contribute to protecting and improving health but are also central to overall human development and the reduction of poverty. Indeed, the current Goal 3 of the Agenda for Sustainable Development 2030 seeks to ensure health and well-being for all, at every stage of life. The target of universal health coverage by 2030 was also adopted as part of the agenda, including financial risk protection, access to quality essential healthcare services and access to safe, effective, quality and affordable essential medicines and vaccines for all.[31] Universal health coverage has therefore become a major goal for health reforms in many countries and strengthening social accountability is a necessary strategy to achieve it.

The preliminary results of our ongoing research collaboration show that health facilities (hospitals, health centres and dispensaries) have now been granted wide decision-making space in the identification of priorities, planning, budgeting and management of financial resources; however, there have been few efforts to strengthen the capacity of the health facilities and the community in order to handle new responsibilities in the context of the current decentralisation. Priority setting, planning and the budgeting process are consequently mainly driven by the healthcare workers, based on historical documents, and rarely informed by the local evidence. Similarly, as reported in earlier studies, there are weak accountability mechanisms at the community and local levels.[9 15 17] While HFGCs are important and can serve as bridges between health facilities and the communities they serve, efforts need to be made to reinforce their capacity; strengthen relationships between HFGCs and the healthcare providers, health managers and community leaders; and sensitise communities to the roles and responsibilities of HFGC members. This project therefore seeks to address the gaps in the HFGCs identified in our previous and ongoing research collaboration.

While the theoretical and participatory elements of the study are a strength, there are several issues that should be considered. Since this is a novel approach and the combination of RE and SSM along the research process is not yet well defined, difficulties could arise when trying to integrate them in one or more steps. These include the challenges of involving multiple stakeholders in defining outcomes as well as in theory development in relation to the complex tasks of HFGCs aiming for social accountability. The contextual factors are layered (micromeso and macro) as well as time dependent, creating a complex landscape for action. Given the uncertainty and complexity of the participatory action research approach combining RE and SSM, continuous adaptations will be required, and the labour-intensive and time-intensive process of the methodologies should not be overlooked.

**Contributors** The Tanzania team (SM, PK, NAK, RK, CM) came with the research idea which was discussed with the Swedish team (MSS, A-KH). MSS and A-KH led the writing of the protocol with support and contributions of the rest of coauthors.

**Funding** This project has been funded by the Vetenskapsrådet (dnr: 2021-04218).

**Competing interests** None declared.

**Patient and public involvement** Patients and/or the public were not involved in the design, or conduct, or reporting, or dissemination plans of this research.

**Patient consent for publication** Not applicable.

**Provenance and peer review** Not commissioned; externally peer reviewed.

**ORCID iDs**
Miguel San Sebastian http://orcid.org/0000-0001-7234-3510
Stephen Maluka http://orcid.org/0000-0002-0369-0858

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
