## [Reviewer comments · BMJ Open]

ARTICLE DETAILS

TITLE (PROVISIONAL)	The role of health facility governing committees in strengthening social accountability to improve the health system in Tanzania: protocol for a participatory action research study
AUTHORS	San Sebastian, Miguel; Maluka, Stephen; Kamuzora, Peter; Kapologwe, Ntuli; Kigume, Ramadhani; Masawe, Cresencia; Hurtig, Anna-Karin

VERSION 1 – REVIEW

REVIEWER	Rifkin, Susan B London School of Hygiene and Tropical Medicine Faculty of Public Health and Policy, Distance learning tutor
REVIEW RETURNED	07-Oct-2022

GENERAL COMMENTS	General Comments: 1. This paper describes a very interesting study protocol to address key issues for health facility governing committees using Tanzania as an example.2. However, for a submission as a research protocol for the investigation, this presentation is light. More detail specifically for the justification for the research focusing on social accountability and for the methodology is necessary.3. The references for this paper are not expansive. For the arguments that are presented for undertaking the study, several of the articles are dated. More publications to support the case have been published in the last two years. A more in-depth discussion about the value of social accountability to address the issues raised to improve the function of the HFGCs would strengthen the paper.4. The presentation of the methodology for examining the value and output of these committees and for making recommendations is not describe in any detail. Examples of its use in specific situations and their value in these situations should be added. Realistic evaluations are complex and challenging. To understand their contribution to this research, a more detailed description is needed. In addition to describing the actual steps and their contribution to outcomes, it is important to highlight the challenges. It would help to give some examples of their use, particularly for assessing participatory evaluations highlighting some of the obstacles particularly in outcomes and evaluations. A similar approach is necessary for the soft systems methodology. Specific Comments: 1. Point 2 on Page 3 about Strengths and Limitation should be rephrased. The outcome you state cannot be assured. I suggest
---

	"To examine whether this approach will strengthen the ability of HFGCs to develop social accountability through participatory learning". 2. P. 6 Some examples of where social accountability have been strengthened through participatory approach would strengthen the paper. 3. P. 10. The choice of districts is not clear. Line 50/51 states there are three categories of districts. However, in Line 45 you say you will choose 2 districts. Please clarify. 4. The English is not always used correctly in the paper. Please review. If the presentation did not have a native English speaker in its preparation, it would be good to ask a native speaker to help.
--	--

REVIEWER	Uddin, Jasim Monash University, Monash
REVIEW RETURNED	24-Nov-2022

GENERAL COMMENTS	The research topic is timely and interesting. However, the paper should be extensively revised to improve its rigor and contribution. Please address the comments below for further improvement.  • It is difficult to follow the paper because of its poor structure and writing style. Please improve writing style in terms of clarity and transition from one to another section. • Literature review on social accountability is under-developed. Please identify a research gap in the extant literature concerning the context of this research. • The method section offers relevant contents on qualitative research. • The findings section is well discussed. • The discussion section is poorly developed. In this section, the authors should list down/point out the list of contributions and each of them contribution should be justified – linking to prior studies. The author should provide implications for policy makers in this section. • The authors should include a concluding remarks section and this section should include limitations of this research and scope for further research within the domain of social accountability. • This paper should conduct proofreading by a professional service provider.
---

VERSION 1 – AUTHOR RESPONSE

Reviewer: 1

Dr. Susan B Rifkin, London School of Hygiene and Tropical Medicine Faculty of Public Health and Policy Comments to the Author:

Comments on "Strengthening social accountability to improve the health system in Tanzania: how can health facility governing committees fulfil their role? A study protocol"

General Comments:

1. This paper describes a very interesting study protocol to address key issues for health facility governing committees using Tanzania as an example.

---Thanks.

2. However, for a submission as a research protocol for the investigation, this presentation is light. More detail specifically for the justification for the research focusing on social accountability and for the methodology is necessary.

--- Thanks for this comment. We have reviewed and rewritten the introduction section highlighting social accountability for health systems and related research. We have updated the references and articulated the research gap. Please see the introduction section with track-changes p 4-6.

The suggested methodology is further justified on p 8 in the introduction to the study design:

“Realist evaluation is a theory-driven approach that focuses on explaining the mechanisms of action that underlie a complex programme or intervention. Just like SSM, realist evaluation is underpinned by complex theory thinking. The SSM has an explicit orientation towards the experiential learning of a group of participants, and therefore adds value to the “method neutral” realist evaluation by engaging stakeholders from the outset. This combination responds to a recent call that realist evaluations should adopt participatory (action) research methods and engage with issues of (power) inequities contributing to “a better understanding of the intervention and may facilitate the emancipation of the disenfranchised” (26). The combination of realist evaluation and SSM allows for inclusion of local stakeholders which will support complementary theory building and theories being rooted in lived realities.”

3. The references for this paper are not expansive. For the arguments that are presented for undertaking the study, several of the articles are dated. More publications to support the case have been published in the last two years. A more in-depth discussion about the value of social accountability to address the issues raised to improve the function of the HFGCs would strengthen the paper.

--- We have reviewed the literature updating references along the manuscript, including the literature regarding the Tanzanian context.

--- We have also rewritten the introduction to highlight the importance of social accountability for health systems, clarifying that the HFGCs is a key social accountability intervention within the Tanzanian health system.

Please see the introduction section with track-changes p 4-6.

4. The presentation of the methodology for examining the value and output of these committees and for making recommendations is not describe in any detail.

Examples of its use in specific situations and their value in these situations should be added. Realistic evaluations are complex and challenging. To understand their contribution to this research, a more detailed description is needed. In addition to describing the actual steps and their contribution to outcomes, it is important to highlight the challenges. It would help to give some examples of their use, particularly for assessing participatory evaluations highlighting some of the obstacles particularly in outcomes and evaluations. A similar approach is necessary for the soft systems methodology.

--- Unfortunately we cannot provide a more detailed description of the methodology under the constraint of 4000 words. We have, however, added a paragraph at the end of the manuscript acknowledging the challenges involved in the application of these methodologies.

While the theoretical and participatory elements of the study are a strength, there are several issues that should be considered. These include the challenges of involving multiple stakeholders in defining outcomes as well as in theory development in relation to the complex tasks of HFGCs aiming for social accountability. The contextual factors are layered (micro-meso and macro) as well as time dependent, creating a complex landscape for action. Given the uncertainty and complexity of the participatory action research combining realist evaluation and SSM, continuous adaptations will be required, and the labour- and time-intensive process of the methodologies should not be overlooked.

Specific Comments:

1. Point 2 on Page 3 about Strengths and Limitation should be rephrased. The outcome you state cannot be assured. I suggest "To examine whether this approach will strengthen the ability of HFGCs to develop social accountability through participatory learning".

--- Thanks, done.

2. P. 6 Some examples of where social accountability have been strengthened through participatory approach would strengthen the paper.

--- Several references have been added and discussed, such as:

6. Danhoundo G, Nasiri K, Wiktorowicz ME. Improving social accountability processes in the health sector in sub-Saharan Africa: a systematic review. *BMC public health*. 2018;18:1-8.

8. Hernandez A, Hurtig A-K, San Sebastian M, Jerez F, Flores W. 'History obligates us to do it': political capabilities of Indigenous grassroots leaders of health accountability initiatives in rural Guatemala. *BMJ Global Health*. 2022;7(5):e008530.

16. Kesale AM, Mahonge C, Muhanga M. The quest for accountability of Health Facility Governing committees implementing direct Health Facility Financing in Tanzania: a supply-side experience. *PLoS one*. 2022;17(4):e0267708.

19. Falisse J-B, Ntakarutimana L. When information is not power: Community-elected health facility committees and health facility performance indicators. *Social Science & Medicine*. 2020;265:113331.

20. Abimbola S, Drabarek D, Molemodile SK. Self-reliance or social accountability? The raison d'être of community health committees in Nigeria. *The International Journal of Health Planning and Management*. 2022;37(3):1722-35.

22. Francetic I, Fink G, Tediosi F. Impact of social accountability monitoring on health facility performance: Evidence from Tanzania. *Health economics*. 2021;30(4):766-85.

3. P. 10. The choice of districts is not clear. Line 50/51 states there are three categories of districts. However, in Line 45 you say you will choose 2 districts. Please clarify.

--- Thanks. This has been corrected, it is two districts which are included.

4. The English is not always used correctly in the paper. Please review. If the presentation did not have a native English speaker in its preparation, it would be good to ask a native speaker to help.

--- The text has now been proof-read.

Reviewer: 2

Dr. Jasim Uddin, Monash University

Comments to the Author:

The research topic is timely and interesting. However, the paper should be extensively revised to improve its rigor and contribution. Please address the comments below for further improvement.

It is difficult to follow the paper because of its poor structure and writing style. Please improve writing style in terms of clarity and transition from one to another section.

---We have thoroughly revised the paper and it has also been proof-read.

Literature review on social accountability is under-developed. Please identify a research gap in the extant literature concerning the context of this research.

--- Thanks, we have now updated the literature. See also response to reviewer 1 (Specific comments, point 2)

A paragraph on the research gap in the end of the introduction section, p 6, has also been added:

“Explorative studies from different contexts have suggested that training (11, 18, 19) is a way forward to empower HFGCs, but they have also highlighted the importance of paying attention to the socio-economic and cultural contexts in which they operate, including power relationships between health care providers and the HFGCs, and political capabilities and how HFGCs perceive their role (8, 19, 20). There is, however, a scarcity of participatory (action) research on how to better strengthen and implement social accountability mechanisms, such as the HFGCs, and their relationships with health care providers and the community they serve (16, 18, 22).”

The method section offers relevant contents on qualitative research.

The findings section is well discussed.

---Thanks.

The discussion section is poorly developed. In this section, the authors should list down/point out the list of contributions and each of them contribution should be justified - linking to prior studies. The author should provide implications for policy makers in this section.

The authors should include a concluding remarks section and this section should include limitations of this research and scope for further research within the domain of social accountability.

--- Please see response to reviewer 1 (point 4) regarding a concluding paragraph. We appreciate the comment of the reviewer wanting a more developed discussion. However, there is no guidelines regarding the content of this section in the journal and as such, we have been guided by the format of previous published protocols in the journal. Additionally, as mentioned in the cover letter, we are constricted by the limit of 4000 words for this type of articles.

This paper should conduct proofreading by a professional service provider.

--- Thanks, this has been done.

VERSION 2 – REVIEW

REVIEWER	Rifkin, Susan B London School of Hygiene and Tropical Medicine Faculty of Public Health and Policy, Distance learning tutor
REVIEW RETURNED	13-Mar-2023

GENERAL COMMENTS	General Comments: 1. The paper is much improved. Details have been addressed. More references to outcomes of the methodology have been added. The English language is much clearer.2. The statement of strengths and limitations only presents strengths. A limitation which you describe later is the challenge of ethics and dissemination.3. I still have a concern about the presentation of the methodology sections. I do understand the challenge with the word limitation. However, this paper focuses on the way in which information will be collected to address issues around social accountability. A clear description of this methodology is needed. The realist evaluation only describes expected outcomes. It does not describe how information will be collected and analyzed. For example ,on p. 13 under, Step 4, it states under step 4 of the realist evaluation. However, the steps of the evaluation are not described in the paper.4. I suggest the following article which presents the approach of realist research in detail with a diagram of the steps taken for data collection and analysis. The challenge of complexity in evaluating health policies and programs: the case of women’s participatory groups to improve antenatal outcomes Van Belle et al. BMC Health Services Research (2017) 17:687 DOI 10.1186/s12913-017-2627-5. It is also necessary to provide the same information for SSM. Without details of how the research is designed and implemented in various stages, it is not clear how the methodology will address the challenges you define.
---

VERSION 2 – AUTHOR RESPONSE

Reviewer: 1

Dr. Susan B Rifkin, London School of Hygiene and Tropical Medicine Faculty of Public Health and Policy Comments to the Author:

Comments on “Strengthening social accountability to improve the health system in Tanzania: how can health facility governing committees fulfil their role? A study protocol”

General Comments:

1. The paper is much improved. Details have been addressed. More references to outcomes of the methodology have been added. The English language is much clearer.

---Thanks.

2.. The statement of strengths and limitations only presents strengths. A limitation which you describe later is the challenge of ethics and dissemination.

--- Thanks. We have now revised the Strengths and limitations section (see above).

3. I still have a concern about the presentation of the methodology sections. I do understand the challenge with the word limitation [NOTE FROM THE EDITORS: PLEASE SEE COMMENT FROM THE EDITORS ABOVE]. However, this paper focuses on the way in which information will be collected to address issues around social accountability. A clear description of this methodology is needed. The realist evaluation only describes expected outcomes. It does not describe how information will be collected and analyzed. For example, on p. 13 under, Step 4, it states under step 4 of the realist evaluation. However, the steps of the evaluation are not described in the paper.

--- Thanks for the comment. We have now clarified that the methodology is following Dalkin et al, 2018 which propose a combination of RE and SSM. The proposal does not follow a classical RE design but a combination with SSM, creating a novel approach as described by Dalkin et al. It is this approach the one that has guided the four different steps of our methods section. These steps do not necessarily correspond with the classical RE steps since they are merge with the SSM tool. We have revised the text to make this clear.

--- For instance, and in a summary way,

Step 1: the output are the maps originated from discussions with stakeholders in a workshop;

Step 2: the initial program theory is developed based on map 1 and represented as map 2 (first step in Van Belle).

Step 3: maps are compared and actions planned based on the needs (this would correspond to the data collection-analysis-synthesis steps of Van Belle process). In this approach this is not done for instance through mixed-methods data collection but through a reflective process supported by specific SSM tools such as matrixes.

Step 4: includes a cycle of several meeting to refine the program theory through continuous revisions of the actions decided in step 3 (this would correspond to the end or refined program of Van Belle).

When introducing the study design, we now start with explaining:

This research project will adopt an action research approach labelled community-based participatory research theory-driven evaluation, combining realist evaluation (24) and soft systems methodology (SSM) (25) as suggested by Dalkin et al (26).

4. I suggest the following article which presents the approach of realist research in detail with a diagram of the steps taken for data collection and analysis:

The challenge of complexity in evaluating health policies and programs: the case of women's participatory groups to improve antenatal outcomes Van Belle et al. BMC Health Services Research (2017) 17:687 DOI 10.1186/s12913-017-2627-

--- We have included a similar reference from van Belle et al (28) in the subsection "Realist evaluation."

5. It is also necessary to provide the same information for SSM. Without details of how the research is designed and implemented in various stages, it is not clear how the methodology will address the challenges you define.

--- We have also included a subheading where the SSM is described. We hope that with the clarification presented above, this issue is elucidated. As highlighted above we have also clarified that this project is building on Dalkin et al (26).

VERSION 3 – REVIEW

REVIEWER	Rifkin, Susan B London School of Hygiene and Tropical Medicine Faculty of Public Health and Policy, Distance learning tutor
REVIEW RETURNED	21-Apr-2023
GENERAL COMMENTS	The paper is now ready for publication. One suggestion, you might want to think of the soft system methodology as the basis for data collection to develop the program theory and its iteration as the basis for the research.